# ANKK1 Is a Wnt/PCP Scaffold Protein for Neural F-ACTIN Assembly

**DOI:** 10.3390/ijms251910705

**Published:** 2024-10-04

**Authors:** Laura Domínguez-Berzosa, Lara Cantarero, María Rodríguez-Sanz, Gemma Tort, Elena Garrido, Johanna Troya-Balseca, María Sáez, Xóchitl Helga Castro-Martínez, Sara Fernandez-Lizarbe, Edurne Urquizu, Enrique Calvo, Juan Antonio López, Tomás Palomo, Francesc Palau, Janet Hoenicka

**Affiliations:** 1Laboratory of Neurogenetics and Molecular Medicine, Center for Genomic Sciences in Medicine, Institut de Recerca Sant Joan de Déu, 08950 Barcelona, Spain; lauradb.biolog@gmail.com (L.D.-B.); lara.cantarero@sjd.es (L.C.); maria.rodriguezsanz@sjd.es (M.R.-S.); gemma.tort@sjd.es (G.T.); jtroyab@gmail.com (J.T.-B.); helgacastromartinez@gmail.com (X.H.C.-M.); edurneurquizu@gmail.com (E.U.); francesc.palau@sjd.es (F.P.); 2Centro de Investigación Biomédica en Red de Enfermedades Raras (CIBERER), Instituto de Salud Carlos III (ISCIII), 08950 Barcelona, Spain; 3Laboratory of Neurosciences, Psychiatry Department, Instituto de Investigación Sanitaria del Hospital Universitario 12 de Octubre, Avda. Andalucía s/n, 28041 Madrid, Spaintomas.palomo@gmail.com (T.P.); 4Centro de Investigación Príncipe Felipe (CIPF), 45012 Valencia, Spain; maria.saezga.92@gmail.com (M.S.); sara.fernandez@uv.es (S.F.-L.); 5Unidad de Proteomica, Centro Nacional de Investigaciones Cardiovasculares (CNIC), 28029 Madrid, Spain; ecalvo@cnic.es (E.C.); jalopez@cnic.es (J.A.L.); 6Centro de Investigación Biomédica en Red de Enfermedades Cardiovasculares (CIBERCV), ISCIII, 28029 Madrid, Spain; 7Centro de Investigación Biomédica en Red de Salud Mental (CIBERSAM), ISCIII, 28041 Madrid, Spain; 8ÚNICAS SJD Center, Hospital Sant Joan de Déu, 08950 Barcelona, Spain; 9Division of Pediatrics, Faculty of Medicine and Health Sciences, University of Barcelona, 08036 Barcelona, Spain

**Keywords:** ANKK1, *Taq*IA, addictions, Wnt/PCP, FARP1, F-ACTIN

## Abstract

The *Taq*IA polymorphism is a marker of both the Ankyrin Repeat and Kinase Domain containing I gene (*ANKK1*) encoding a RIP-kinase, and the *DRD2* gene for the dopamine receptor D2. Despite a large number of studies of *Taq*IA in addictions and other psychiatric disorders, there is difficulty in interpreting this genetic phenomenon due to the lack of knowledge about ANKK1 function. In SH-SY5Y neuroblastoma models, we show that ANKK1 interacts with the synapse protein FERM ARH/RhoGEF and Pleckstrin Domain 1 (FARP1), which is a guanine nucleotide exchange factor (GEF) of the RhoGTPases RAC1 and RhoA. ANKK1–FARP1 colocalized in F-ACTIN-rich structures for neuronal maturation and migration, and both proteins activate the Wnt/PCP pathway. ANKK1, but not FARP1, promotes neuritogenesis, and both proteins are involved in neuritic spine outgrowth. Notably, the knockdown of *ANKK1* or *FARP1* affects RhoGTPases expression and neural differentiation. Additionally, ANKK1 binds WGEF, another GEF of Wnt/PCP, regulating its interaction with RhoA. During neuronal differentiation, ANKK1–WGEF interaction is downregulated, while ANKK1–FARP1 interaction is increased, suggesting that ANKK1 recruits Wnt/PCP components for bidirectional control of F-ACTIN assembly. Our results suggest a brain structural basis in *Taq*IA-associated phenotypes.

## 1. Introduction

Addictions are psychiatric disorders characterized by dysregulation of motivational circuits, reward deficits, and modification of learning systems in the brain [1]. Twin studies [2], recurrence in families [3,4], and inherited alterations in brain structure [5,6] support that the risk of developing substance-use disorders has an important hereditary component.

The *Taq*IA single nucleotide variant (SNV, rs1800497) located in the Ankyrin Repeat and Kinase domain containing 1 gene (*ANKK1*) [7] has been associated with addictions in a huge number of studies [8]. *Taq*IA consists of a C/T change producing two alleles referred to as A2 (C) and A1 (T). Blum et al. first reported the association between alcoholism and the *Taq*IA A1 allele and the A1+ genotype (hetero- or homozygous for A1) [9]. As *ANKK1* is adjacent to the *DRD2* gene for the dopamine D2 receptor and the *Taq*IA is a marker of both genes [10], it is difficult to interpret the *Taq*IA-association genetic phenomenon [8,11,12]. We previously found *ANKK1* association with the dopaminergic system in a series of patients with antisocial alcoholism [13], in human peripheral blood mononuclear cells [8] and cellular models [10,14], and in the mouse brain [15]. In zebrafish and mouse models, the link of *Ankk1* with the dopaminergic system has been reported [16,17]. Additionally, ANKK1 localizes to radial glia and other neural progenitors, suggesting its participation in shaping brain structure [10,18]. However, the precise function of ANKK1 is still unknown.

ANKK1 belongs to the Receptor Interacting Protein Kinases (RIPKs) family that share homologous kinase domains with unusual flexibility and conformation roles that extend beyond catalytic functions [19]. RIPKs participate in cell death [20], intracellular inflammatory signaling [21], and/or cellular differentiation [22]. ANKK1 is the RIPK5, which contains 765 amino acids, including an N-terminal RIP kinase and a C-terminal ankyrin repeats domain where the *Taq*IA SNV produces a Glu713-to-Lys (p.Glu713Lys) substitution [7]. RIPK4, which shares the RIP kinase and ankyrin repeats domain with ANKK1, is needed for skin development during embryogenesis and normal skin tissue epidermal differentiation in adults [23].

In this study, we show that ANKK1 is a binding partner of FARP1 (FERM, ARH/RhoGEF, and Pleckstrin Domain Protein 1), a guanine nucleotide exchange factor (GEF) of RhoGTPases [24]. FARP1 is involved in neural development through interactions with cell surface proteins and activation of RhoGTPases regulating dendritic filopodia and spine morphology [24], axon guidance and extension, dendritic complexity, and the number of spines [25,26]. Given that both *ANKK1* and *FARP1* are expressed during prenatal and postnatal neurodevelopment [10,18,25], we hypothesize that ANKK1–FARP1 interaction participates in neurogenic pathways, where regulation of FARP1 activity by ANKK1 plays a key role. We have discovered in SH-SY5Y cell models, both as proliferative neuroblasts and during differentiation as neuron-like phenotype, the function of ANKK1 as a scaffold protein in the Wnt/PCP pathway for neural F-ACTIN assembly.

## 2. Results

### 2.1. ANKK1 and FARP1 Interact in Neural Cells and Colocalize in Neuritic Structures

To identify ANKK1 interactors, we overexpressed the short isoform of ANKK1 fused with the Green Fluorescence Protein (GFP) [14] in HEK293T cells. This isoform comprises only the ANKK1 kinase domain (ANKK1^Thr239^, Appendix A) and is expressed in embryonic/proliferative neurogenesis stages [18]. After co-immunoprecipitation (co-IP) with α-GFP, the analysis by isobaric tagging for relative and absolute quantitation (iTRAQ) mass spectrometry identified candidate proteins that increased in the ANKK1 sample compared with the MOCK control (overexpresses GFP). We selected FARP1 for further analysis because of its high ANKK1/MOCK iTRAQ ratio and biological function in neurons [24,25]. After overexpression of the full-length ANKK1^Thr239^ (fl-ANKK1^Thr239^) and FARP1 constructs (Appendix A and Appendix A), co-IP experiments confirmed ANKK1–FARP1 interaction (Figure 1A). In the neuroblastoma cells SH-SY5Y, co-IP of endogenous ANKK1 and FARP1 demonstrated their interaction under proliferative (neuroblast-like) or serum-starved conditions (0% fetal bovine serum, FBS) that induce neural differentiation (neuronal-like) (Figure 1B). ANKK1–FARP1 interaction was further confirmed by proximity ligation assay (PLA) in undifferentiated cells (Figure 1C) and after retinoic acid (RA) treatment or starvation to induce differentiation (Figure 1D). We observed in differentiated SH-SY5Y cells a significant increase in both the ANKK1–FARP1 interaction and the mRNA expression of *ANKK1* and *FARP1* (Figure 1D,E).

To find out shared functions of ANKK1 and FARP1 during neural differentiation, we generated modified SH-SY5Y cell lines that stably express (stable cell, sc) the short isoform of ANKK1 (scANKK1^Thr239^), the full-length ANKK1 (scfl-ANKK1^Thr239^) or FARP1 (scFARP1) (Appendix A). The study of these undifferentiated cell lines revealed differences in the subcellular localization of both proteins. scfl-ANKK1^Thr239^ presented a greater tendency to neuritic outgrow and significantly higher mRNA levels of Microtubule-associated protein 2 (*MAP2*) mRNA, a neural marker of neuritogenesis (Figure 1G). The scFARP1 cells exhibited larger somas and FARP1 across the entire plasma membrane and lower *MAP2* expression (Figure 1F,G). In addition, after differentiation of the scMOCK^V5^ cells, we found a significant increase in ANKK1 and FARP1 signals (Figure 1F right). These results argued that overexpression of ANKK1 or FARP1 promotes different morphological changes in SH-SY5Y cell lines.

Next, we developed a custom automatic image analysis method to quantify the neurites-to-soma area ratio as a marker of neuritogenesis (Appendix A). Undifferentiated cells showed a significant increase in neurite/soma area in the scANKK1^Thr239^ and scfl-ANKK1^Thr239^ cell lines but not in scFARP1 (Figure 2A). It should be noted that neuritogenesis was increased in the scFARP1 cells after transfection with ANKK1^Thr239^ or fl-ANKK1^Thr239^ plasmids (Figure 2A right), thus suggesting a regulatory role of ANKK1 on FARP1 function. Although no differences were observed between differentiated SH-SY5Y cell lines (Figure 2A, 0% FBS), the more detailed examination of neurites by deconvoluted confocal images using a custom automatic image analysis (Appendix A) revealed an increase in the dendritic spines in scANKK1^Thr239^, scfl-ANKK1^Thr239^, and scFARP1 lines (Figure 2B left). In contrast, the spine length was exclusively associated with lines stably expressing ANKK1 or by transfection of ANKK1 isoforms in the _sc_FARP1 line (Figure 2B right). These results suggest that ANKK1 and FARP1 appear to have synergistic roles in spine formation.

Since FARP1 promotes F-ACTIN assembly [24,27], we immunostained endogenous ANKK1/FARP1, or ANKK1/F-ACTIN (Phalloidin staining) in SH-SY5Y cells treated with RA, or RA and brain-derived neurotrophic factor (BDNF) to obtain mature neurons. We found complete ANKK1–FARP1 and ANKK1–F-ACTIN colocalization signals in the growth cone, neurites, and dendritic filopodia (Appendix A). These findings suggest shared ANKK1/FARP1 functions associated with neuronal maturation.

### 2.2. ANKK1 and FARP1 Participate in Neural Cell Migration

Since ANKK1 is found in migratory neuroblasts and myoblasts [18,28] and FARP1 promotes cell motility [29,30], we analyzed endogenous ANKK1 in subcellular cell migration structures in proliferative SH-SY5Y cells. Immunolabeling experiments showed that ANKK1 was located at the leading edge of F-ACTIN-rich sites, which are responsible for protrusive movements, such as lamellipodia, filopodia, and membrane ruffles (Figure 3A). We did not observe ANKK1 in the lamellum, a less dense F-ACTIN site related to substrate adhesion and contraction. These results suggest that ANKK1 is specifically involved in the forward movement in cell migration.

After, we performed scratch wound healing assays in SH-SY5Y cells transfected with ANKK1^Thr239^, fl-ANKK1^Thr239^, or FARP1 constructs and with the mutant constructs ANKK1^Arg51^ and fl-ANKK1^Arg51^ [p.Lys51Arg change, which would affect kinase activity [14]] (Appendix A and Appendix A). A custom automatic image analysis method was developed to evaluate the wound area (Appendix A). Overexpression of all constructs caused a significant reduction of the wounding area compared to the MOCK at 45 h after scratching. The p.Lys51Arg variant does not affect this reduction, as shown by ANKK1^Arg51^ and fl-ANKK1^Arg51^ constructs (Figure 3B). Since these experiments involved transient transfections, we confirmed that migrating cells closing the gap overexpressed ANKK1 or FARP1 (Appendix A). Moreover, co-transfection of ANKK1 and UtrCH-cherry plasmid for F-ACTIN imaging [31] confirmed that living migrating cells overexpressing ANKK1 were localized at the leading front (Appendix A). These results add further evidence of ANKK1 and FARP1 role in the migration of neural lineage cells and F-ACTIN assembling.

### 2.3. ANKK1 and FARP1 Activate the Non-Canonical Wnt/PCP Pathway

As ANKK1 and FARP1 were involved in neural migration and differentiation, processes regulated by the Wnt Planar Cell Polarity (Wnt/PCP) [32], we hypothesized that both proteins are components of this pathway. We studied the effect of ANKK1^Thr239^ or FARP1 overexpression on the Wnt/PCP pathway with luciferase reporter assays (LRA) that measure Activator Protein-1 (AP-1)-dependent transcriptional activity. In HEK293T cells, we found that the two proteins drove significant AP-1 activity (Figure 4A,B). Since the *Taq*IA *ANKK1* SNV, associated with addictions [8], is a marker of amino acid changes in the ANKK1 proteins, we wondered if this SNV affects the activation of Wnt/PCP induced by ANKK1. We overexpressed ANKK1^Thr239^ (Thr239 allele) and fl-ANKK1^Thr239^ (H1 haplotype, allele Thr239-Lys713) constructs and the ANKK1^Ala239^ (Ala239 allele) and fl-ANKK1^Ala239^ (H2 haplotype, Ala239-Glu713 allele) polymorphic variants (Appendix A and Appendix A). We found that ANKK1^Thr239^ drove a significantly lower AP-1 activity in comparison with ANKK1^Ala239^, while no differences were found between fl-ANKK1^Thr239^ and fl-ANKK1^Ala239^ (Figure 4C).

Next, we studied whether ANKK1 activates the Wnt/PCP pathway in the neural lineage. In undifferentiated SH-SY5Y cells, overexpression of ANKK1^Thr239^ and fl-ANKK1^Thr239^ proteins significantly increased AP-1 activity (Figure 4D). No differences between polymorphic variants were found in this cell line. Furthermore, we examined the temporal activity of Wnt/PCP induction by ANKK1^Thr239^, ANKK1^Arg51^, fl-ANKK1^Thr239^, and fl-ANKK1^Arg51^ constructs in transfected SH-SY5Y cells during differentiation. The expression of constructs was monitored by Western blot (Appendix A). In differentiated SH-SY5Y cells, LRA showed that overexpression of all ANKK1 isoforms significantly increased AP-1 activity compared to proliferative cells. The fl-ANKK1^Thr239^ isoform [expressed during the establishment of neuronal connectivity in early postnatal stages [18]] and fl-ANKK1^Arg51^ caused the highest increase in AP-1 activation (10×) in comparison to the short ANKK1^Thr239^ isoform (Figure 4E).

As the Wnt/PCP pathway activates the Rho family of small GTPases to remodel the ACTIN cytoskeleton [33], we evaluated the interaction of ANKK1 with the RhoGTPases RAC1, RhoA, and Cdc42. In undifferentiated SH-SY5Y lines, co-IP experiments revealed a robust interaction between fl-ANKK1^Thr239^ and RAC1 (Appendix A), while the interaction with RhoA was minimal (Appendix A left) and absent with Cdc42 (Appendix A left). Notably, the mutant isoform scfl-ANKK1^Arg51^ interacted with RAC1 (Appendix A), in agreement with the Wnt/PCP activation by this isoform (Figure 4E). After SH-SY5Y differentiation, our experiments revealed the interaction of fl-ANKK1^Thr239^ with RhoA while interaction with Cdc42 remained absent (Appendix A). PLA experiments confirmed these findings (Appendix A–F) and showed a significant increase in fl-ANKK1^Thr239^–RAC1 and fl-ANKK1^Thr239^–RhoA interactions in differentiated cells. These results argue that ANKK1 and FARP1 are Wnt/PCP components during neural differentiation.

### 2.4. ANKK1 Regulates GEFs/RhoGTPases Interactions during Neural Differentiation

Since ANKK1 interacts with FARP1 and with RAC1 and RhoA, which are activated by FARP1 [24,27], we hypothesized a functional relationship between all these proteins. To assess this possibility, we established ANKK1 and FARP1 shRNA-mediated knockdown SH-SY5Y cell lines named shANKK1 and shFARP1, respectively (Appendix A). The knockdown cells were characterized by qRT-PCR, Western blot, and PLA (Appendix A). qRT-PCR experiments showed coordinated expression of ANKK1 and FARP1 in both cell lines. Under proliferative conditions, we found a significant decrease in the target gene expression and a significant increment of the other (Appendix A left). In contrast, in differentiated cells, the decreased expression of one gene caused a significantly reduced expression of the other (Appendix A right). Moreover, using ENCODE data we identified common transcription factor binding sites in ANKK1 and FARP1 regulatory regions (Appendix A) associated with neural differentiation (Appendix A). This data set added further evidence for functional coordination between these genes. Western blot and PLA showed the reduction of the protein of the target gene (Appendix A).

In the shANKK1 and shFARP1 knockdown lines, we first analyzed RAC1 and RhoA levels by Western blotting. In the shMOCK cells, serum-starved conditions (0% FBS) to induce differentiation caused a significant increase in RAC1 and RhoA. In contrast, the shANKK1 or shFARP1 lines showed no increase of these proteins (Figure 4F). These results suggest that ANKK1 and FARP1 are needed for RAC1 and RhoA participation in neuronal differentiation.

To investigate whether the interaction of ANKK1 with RAC1 and/or RhoA depends on FARP1 and vice versa, we conducted PLA studies in both undifferentiated cells and after 0% FBS treatment to induce neural differentiation. The ANKK1–RAC1 interaction, which increases during differentiation, decreases with the knockdown of *FARP1* (shFARP1) (Figure 4G left). The FARP1–RAC1 interaction remained unchanged in 0% FBS but decreased with the knockdown of *ANKK1* (shANKK1) in undifferentiated cells. However, shANKK1 in 0% FBS showed a significant increment of FARP1–RAC1 interaction, suggesting an ANKK1 regulatory role in this interaction (Figure 4G right). Since the ANKK1–FARP1 interaction increased during differentiation (Figure 1D), these results suggest a functional relationship between ANKK1–FARP1–RAC1 necessary for RAC1 activation during neuronal differentiation. Regarding the ANKK1–FARP1–RhoA relationship, we found that the ANKK1–RhoA interaction is independent of FARP1. In contrast, the FARP1–RhoA interaction was greater in proliferation, decreased during differentiation, and required ANKK1 (Figure 4H). These results suggest that ANKK1 facilitates interactions between FARP1 and their substrates.

Next, we studied the cellular morphology and F-ACTIN pattern in shANKK1 cells treated with RA and BDNF to obtain mature neurons. The undifferentiated knockdown of *ANKK1* (shANKK1) showed slight morphological differences with a tendency to a lower number of neurites and a larger soma area compared to the shMOCK, although with similar levels of F-ACTIN (Figure 5A left). After the RA and RA/BDNF treatment, the shMOCK line significantly changed its morphology to a neuronal-like phenotype (Figure 5A right). In contrast, the shANKK1 line did not undergo differentiation and retained its proliferative morphology (Figure 5A), significantly decreasing the F-ACTIN signal compared to shMOCK (Figure 5B,C). Qualitatively, the F-ACTIN in RA- and RA/BDNF-treated shANKK1 cells showed a stress fibers pattern typically associated with RhoA activity. These results indicate that without ANKK1, the necessary F-ACTIN assembly for differentiation into a neural phenotype does not occur.

### 2.5. ANKK1 Works as a Scaffold Protein Regulating GEFs/RhoGTPases Interactions

We searched among possible interactors to study whether ANKK1 interacts with other GEFs. We found RhoGEF19 (WGEF), which has been previously reported to activate RhoA in the Wnt/PCP [34]. Co-IP and PLA experiments confirmed ANKK1–WGEF interaction (Figure 6A,B). This interaction decreased significantly in differentiated cells (Figure 6B). It is noteworthy that RhoA, a WGEF substrate, showed a significant increase in its interaction with ANKK1 during differentiation (Figure 4H). Thus, we propose an inverse relationship between ANKK1–RhoA and ANKK1–WGEF interactions during SH-SY5Y differentiation. Furthermore, the WGEF–RhoA interaction increased when ANKK1 was knocked down (shANKK1) (Figure 6C), suggesting that ANKK1 regulates this interaction. The immunostaining of WGEF in proliferative shMOCK and shANKK1 cells showed a broad cellular distribution, while after differentiation, we observed a significant decrease in the WGEF signal in shANKK1 (Figure 6D). In addition, we studied ANKK1 and WGEF colocalization in the scfl-ANKK1^Thr239^ differentiated cell line using STED microscopy. Although both proteins localize to neurites, colocalization was low (Appendix A), consistent with a decrease in ANKK1–WGEF interaction during differentiation. Altogether, these results suggest that ANKK1 is required to downregulate the WGEF–RhoA interaction during SH-SY5Y neural differentiation.

**Figure 5 ijms-25-10705-f005:**
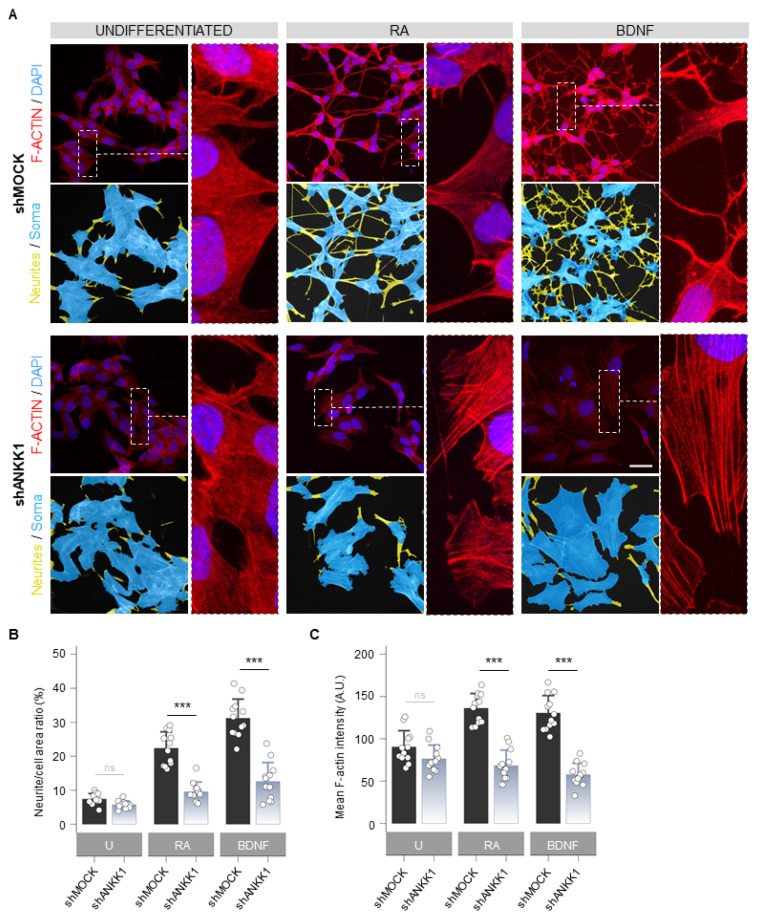
*ANKK1* knockdown prevents neural differentiation and F-ACTIN assembly. (**A**). F-ACTIN staining (phalloidin) in undifferentiated (U) and RA or BDNF-treated shMOCK and shANKK1 knockdown cell lines. The corresponding neurites (yellow)/soma (blue) segmentation results are displayed below each original image. A magnification detail of F-ACTIN staining is also shown (image contrast was equalized to appreciate F-ACTIN fibers pattern). Maximum intensity projections are shown. Scale bar: 30 µm. (**B**) Bar plots showing neurite (yellow)/cell (yellow + blue) area ratio as a marker of neuritogenesis (n = 12 images, 3 independent experiments). (**C**) Bar plots showing F-ACTIN fluorescence intensity quantification (A.U.) as a marker of F-ACTIN polymerization (n = 12 images in each category, 3 independent experiments). Plots represent mean ± SD and individual values are displayed as dots. Student’s *t*-test was used for MOCK-group comparisons. Bonferroni was used as a multiple-testing correction method. *** *p* < 0.001; ns, not significant. Abbreviations: A.U.: Arbitrary units; BDNF: Brain-derived neurotrophic factor; RA: Retinoic acid; U: Undifferentiated.

ANKK1 interactions with the Wnt/PCP components FARP1, WGEF, RAC1, and RhoA lead us to propose that ANKK1 functions as a scaffold protein within this pathway. Considering this, we obtained the ANKK1 structure from the AlphaFold database [35,36] (Uniprot Q8NFD2) to perform in silico studies (Figure 6E). In addition to the RIP kinase and C-terminal ankyrin repeats domains, ANKK1 contains five polymorphic amino acid changes in its sequence that generate specific haplotypes (H2B, H2, and H1). The H2B haplotype is common to all populations, the H2 is the most common in Caucasians, and the H1 includes the risk allele A1 for the *Taq*IA SNV (Appendix A). The in silico study using PyMOL (version 1.8) [37] revealed no differences in structure, folding, or flexibility between these ANKK1 haplotypes. However, ANKK1 showed a high B-factor indicative of high flexibility in the RIP kinase and the ankyrin repeats domain (Figure 6F). B-factor for ANKK1 ranges from 26.85 to 98.63 Å2, with a mean value of 82.75 Å2. The largest B-factors are located between the α21–α39 helices of the Ankyrin repeats domain (87.2–98.63 Å2). The kinase domain also shows elevated B-factors for the α5, α6, α10, and α13 helices (90.02–98.06 Å2). Additionally, the electrostatic mapping showed significant differences in the ankyrin repeats domain between ANKK1 haplotypes involving the *Taq*IA SNV position (Figure 6G). We found changes in the ANKK1 surface for each haplotype/allele in the amino acids 318 [Arg(+) in H2 and Gly(n) in H2B/H1]; 442 [Arg(+) in H2/H1 and Gly(n) in H2B]; 490 [His(+) in H2B/H1 and Arg(+) in H2] and 713 [Glu(−) in H2B/H2 and Lys(+) in H1]. Moreover, we found 56 putative binding sites (Appendix A), which could facilitate protein interactions and scaffolding properties.

## 3. Discussion

ANKK1 has been widely associated with addictions, learning paradigms, and other dopaminergic-related phenotypes, especially its *Taq*IA SNV [8]. Yet, the function of ANKK1 was unknown, and the biological basis of this genetic phenomenon remained unclear. This work shows that ANKK1 is required to modulate the classical GEFs FARP1 and WGEF in their interaction with their substrates RAC1/RhoA and RhoA, respectively. Inferred from these ANKK1 interactors, and considering that ANKK1 overexpression activates the Wnt/PCP signaling, we propose that ANKK1 is a scaffold protein of Wnt/PCP that recruits RhoGEFs in regulating RhoGTPases activation in proliferative neuroblasts and during their differentiation into neurons.

This proposal is based on the following lines of evidence. First, ANKK1 and FARP1 interact at different molecular and cellular levels: (i) genetically, since their expression is coordinated in neuroblasts and during neuronal differentiation; (ii) protein-wise, since ANKK1 and FARP1 interact and this interaction increases during neuronal differentiation; (iii) within the Wnt/PCP pathway, which is activated by individual overexpression of both proteins; (iv) in their cellular function, since both promote morphological changes in the neural lineage, leading the differentiation and maturation of neurons; (v) in their subcellular location, since they colocalize within the plasmatic membrane and in F-ACTIN rich structures for neural maturation and migration; and finally, (vi) both proteins are necessary for RAC1 and RhoA expression and F-ACTIN assembly during neural differentiation. Second, ANKK1 interacts with WGEF, which activates RhoA in the Wnt/PCP pathway [34]. During neuronal differentiation, the ANKK1–WGEF interaction is downregulated, whereas the ANKK1–FARP1 interaction is increased, suggesting that ANKK1 is required to recruit components of Wnt/PCP signaling for the bidirectional control of F-ACTIN assembly.

Given that FARP1 is involved in dendritic growth and synapsis [24,38,39], the ANKK1–FARP1–RAC1 functional relationship may play a role during the development of brain structure or promote neuroplasticity events underlying *ANKK1*-associated conditions [8]. On the other hand, WGEF belongs to the Ephexin GEFs subfamily involved in neuron development and synaptic homeostasis via F-ACTIN reorganization [40] and regulates planar cell polarity in vertebrate brain development [41]. On the ANKK1 side, our results align with the neurodevelopmental-related locus on chromosome 11q22-23, where *ANKK1* maps [42]. In previous work, we demonstrated in mice that the *Ankk1* gene and proteins have their maximum expression during embryonic and postnatal development and localize in radial glia and other progenitors [10,18]. Furthermore, ANKK1 is increased in mitotic neural progenitors [18], and Wnt/PCP has been implicated in asymmetric mitosis of neural progenitors during development [43]. Here, we demonstrate that ANKK1 is a component of Wnt/PCP participating in different stages of neural development directed by this pathway, such as neuritogenesis, dendritogenesis, and migration of neural precursors [44]. Our results are consistent with the location of ANKK1 within neural precursors [10,18] and suggest that this protein is relevant to the structural conformation of the brain.

From a clinical point of view, our results would offer a molecular explanation for the association of heritable differences in both structure [5] and cortico-striatal connectivity [6] of the human brain and vulnerability to addiction. Considering that lower cortico-striatal connectivity is associated with vulnerability to addictions [6], we propose that carriers of the A2 allele (H2B and H2 haplotypes) would have a greater cortico-striatal connection than carriers of the A1 allele who are predisposed to addictions. At the neurocognitive level, the A1 allele has been associated with midbrain structures related to addiction biology [45,46,47] and reduced activity in the prefrontal cortex and striatum [48]. Additionally, carriers of the A1 risk allele show greater impulsivity [49,50,51,52,53] and deficiencies in working memory [54], reverse learning [48], learning from errors [55], and long-term memory [56]. All of these deficiencies cause a greater predisposition to addictions.

The genetic phenomenon of association with the *Taq*IA *ANKK1* SNV, however, goes beyond addictions to other psychiatric illnesses such as schizophrenia, eating disorders, and some childhood behavior disorders [57]. Given the relationship between ANKK1 and Wnt/PCP, all of these *Taq*IA-associated phenotypes could have a basis in brain structure for their expression [57]. Moreover, Wnt pathways are involved in ventral midbrain neurogenesis and dopaminergic axon morphogenesis during development [58,59,60] and also in adult neurogenesis [61,62,63]. Differences in ANKK1 scaffolding properties related to *ANKK1* haplotypes and their impact on Wnt/PCP are relevant in these scenarios. If this is the case, *ANKK1* polymorphic alleles could potentially impact prenatal Wnt/PCP during brain development, which is a plausible biological mechanism underlying the association between the *Taq*IA, the psychiatric phenotypes, and human brain structure. Future studies of the WNT/PCP pathway in *ANKK1* knockout mouse models are necessary to provide more information about the involvement of ANKK1 proteins and their influence on behavior and susceptibility to addiction.

In conclusion, here we provide the first evidence for the precise function of ANKK1 and a biological correlate to the wide range of behavioral phenotypes and psychiatric disorders associated with the *ANKK1 Taq*IA SNV. Henceforward, the *Taq*IA-associated phenotypes can no longer be explained only by functional variations in the dopamine receptor D2, as our findings support a definitive role for ANKK1. There is still work to be performed to understand the implications of the ANKK1 alleles and their relationship to tissue- or context-dependent components of Wnt/PCP and their functions. This knowledge could reveal how brain structural differences underlie vulnerability or resilience to changes induced either by drug use in the case of addiction or by environmental factors that trigger other ANKK1-associated disorders or behaviors.

## 4. Materials and Methods

### 4.1. Study Design, Materials and Data Collection

This study was designed to identify ANKK1 pathways and interactors using as a first step, isobaric tagging for relative and absolute quantitation (iTRAQ) mass spectrometry of protein extracts from HEK293 cells overexpressing ANKK1. ANKK1–FARP1 interaction was characterized in SH-SY5Y cells with neuroblast-like morphology, which are positive for tyrosine hydroxylase (TH) and dopamine-β-hydroxylase and can be differentiated to a mature neuron-like phenotype characterized by neuronal markers [64]. We generated modified SH-SY5Y cell lines that stably express ANKK1 or FARP1, as well as knockdown cell lines for these proteins. The cellular models were studied under proliferative conditions (undifferentiated) or after treatment with retinoic acid (RA) or serum-starved conditions to induce their differentiation into neurons. A detailed description of the materials and methods can be found as Appendix A.

### 4.2. Statistical Analysis

Statistical analysis was performed using R software for Windows (version 4.3.1). The Shapiro–Wilk test was used to check the normality assumption for data distribution. When necessary, we normalized data relative to controls to mitigate variability between assays. In these cases, we performed a one-sample *t*-test (≠1) for each groups compared to the control (forced to be =1 because of normalization), and Student’s *t-*test (2 groups) or ANOVA (>2 groups) for each group-to-group comparison. For the remaining analyses, we used Student’s *t-*test (2 groups) or ANOVA (>2 groups) for each group-to-group comparison. To reduce the risk of false discoveries stemming from multiple tests, we employed the Bonferroni correction.

We developed custom automatic image analysis methods to quantify and qualify neural structures (Appendix A). An independent investigator performed the analysis of the experimental results blindly. Sample sizes were estimated based on our previous experience with each technique. Outliers were excluded from the analysis. All experiments with biological replicates are indicated in the figure legends. The details of the materials and methods are in the Appendix A. To evaluate wound area reduction over time, we performed mixed-effects linear regression. All statistical tests defined *p* < 0.05 as significant. Unless otherwise specified, all results are expressed as mean ± 1 SD. The experiments were performed at least three times.

## Figures and Tables

**Figure 1 ijms-25-10705-f001:**
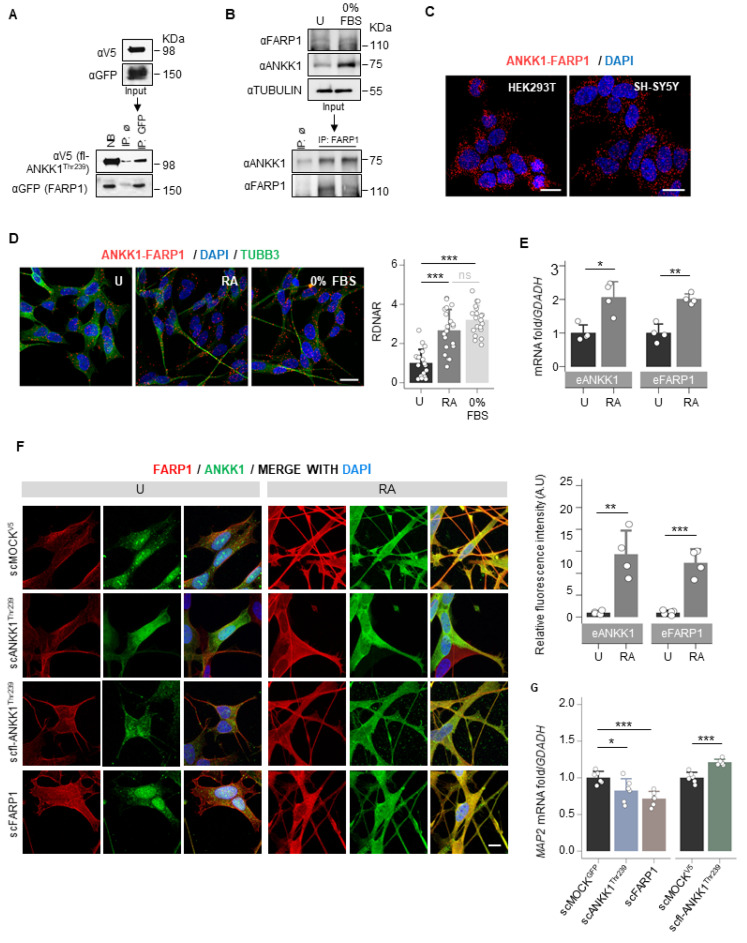
ANKK1 and FARP1 interact constitutively and during neural differentiation. (**A**). Co-IP of transfected FARP1 and fl-ANKK1^Thr239^ in HEK293T cells. (**B**). Co-IP of endogenous FARP1 and ANKK1 in proliferative and 48 h serum-starved SH-SY5Y cells (0% FBS). (**C**,**D**). ANKK1–FARP1 interaction by PLA in HEK293T or SH-SY5Y. In SH-SY5Y, PLA was combined with immunostaining of TUBB3 in undifferentiated, and RA- or 0% FBS-treated cells. Maximum intensity projections are shown. Scale bar: 20 µm. Analysis of RDNAR is shown in the bar plot (n = 20 images, 3 independent experiments). (**E**). Quantification by RT-PCR of ANKK1 and FARP1 expression levels in undifferentiated and RA-treated SH-SY5Y cells (n = 3 independent experiments). (**F**). Immunolabelling of ANKK1 and FARP1 in undifferentiated or RA-treatment conditions of the following SH-SY5Y cell lines that stably express ANKK1 or FARP1: scMOCK^V5^ (α-ANKK1 (StK2) and α-FARP1); scANKK1^Thr239^ for the short isoform in fusion with GFP (α-GFP and α-FARP1); scfl-ANKK1^Thr239^ for the full-length isoform in fusion with V5 (α-V5 and α-FARP1) and scFARP1 for FARP1 in fusion with GFP (α-ANKK1 (StK2) and α-GFP). Maximum intensity projections are shown. Scale bar: 10 µm**.** Quantification of ANKK1 and FARP1 fluorescence intensity is shown in the bar plots (n = 3 independent experiments). (**G**). qRT-PCR of Microtubule-associated protein 2 gene (*MAP2*) in undifferentiated scANKK1, scFARP1, and scfl-ANKK1^Thr239^ cell lines. Plots represent mean ± SD and individual values are displayed as dots. One-sample *t*-test on the fold-change values was used for U-group comparisons. Bonferroni was used as a multiple-testing correction method. * *p* < 0.05; ** *p* < 0.01; *** *p* < 0.001; ns, not significant. Abbreviations: FBS: Fetal Bovine Serum; IP: immunoprecipitation; NB: not-bound; IP ø: non-specific IP; PLA: Proximity ligation assay; RA: Retinoic acid. See also Appendix A.

**Figure 2 ijms-25-10705-f002:**
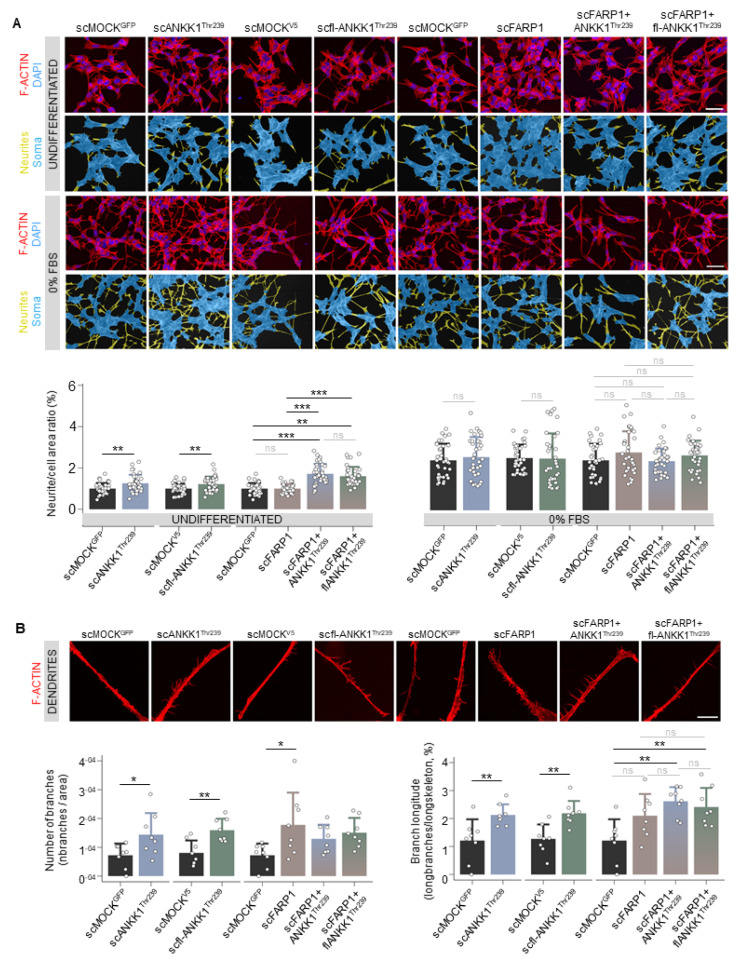
ANKK1 but not FARP1 promotes neuritogenesis, and both proteins contribute to the growth of neuritic spines. (**A**). F-ACTIN (phalloidin) staining of undifferentiated (U) or 48 h 0% FBS-treated scMOCK, scANKK1^Thr239^, scfl-ANKK1^Thr239^, scFARP1 cell lines, and scFARP1 transfected with ANKK1^Thr239^ or fl-ANKK1^Thr239^ constructs. Maximum intensity projections are shown. Scale bar: 40 µm. The corresponding neurites (yellow)/soma (blue) image segmentation are displayed below each original image. Analysis of neurite/cell area ratio (as a marker of differentiation) in undifferentiated (upper panel) and after 48 h of 0% FBS (lower panel) is shown in the bar plots below (n = 35 images, 3 independent experiments). (**B**). Representative deconvoluted confocal images of dendritic spines after 48 h of 0% FBS. Maximum intensity projections are shown. Scale bar: 10 µm. Analysis of the relative number of branches (left panel) and branch longitude (right panel) as markers of dendritogenesis is shown in the bar plots below (n = 8 images, 3 independent experiments). Plots represent mean ± SD and individual values are displayed as dots. (**A**,**B**) One-sample *t*-test on the fold-change values was used for each MOCK-group comparison and Bonferroni was applied as a multiple-testing correction method. ANOVA on the fold-change values was used for between-group comparison. * *p* < 0.05; ** *p* < 0.01; *** *p* < 0.001; ns, not significant. Abbreviations: FBS: Fetal bovine serum; U: Undifferentiated. See also Appendix A.

**Figure 3 ijms-25-10705-f003:**
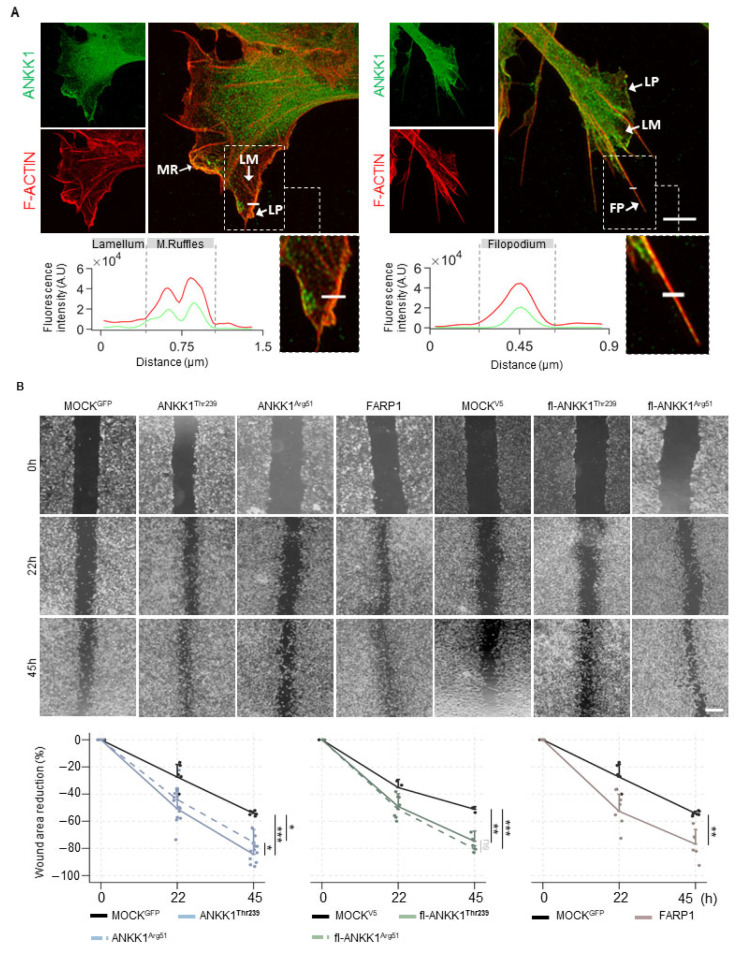
ANKK1 and FARP1 are involved in neural cell migration. (**A**). Representative confocal images showing endogenous ANKK1 (α-STk2) and F-ACTIN (Phalloidin) at subcellular cell migration structures (white arrows) of undifferentiated SH-SY5Y cells. Maximum intensity projections are shown. Scale bar: 2 µm. The line profile plots indicate the intensity distribution of the green and red channels along the straight line crossing the ROI (magnified boxes). (**B**). Representative microscopy images of the scratch wound healing assay at 0, 22, and 45 h in SH-SY5Y cells transiently overexpressing ANKK1 (n = 8 independent experiments), ANKK1^Arg51^ (n = 8 independent experiments), FARP1 (n = 8 independent experiments), fl-ANKK1^Thr239^ (n = 3 independent experiments), and fl-ANKK1^Arg51^ (n = 3 independent experiments). Scale bar: 500 µm. Line plots represent the percent rates of wounded areas relative to the start time of scratch (mean ± SD). Individual values are displayed as dots. Mixed effects linear regression was used to estimate wound healing mean change over time across the different cell groups. * *p* < 0.05; ** *p* < 0.01; *** *p* < 0.001; ns, not significant. Abbreviations: A.U: Arbitrary units; FP (Filopodium); LM (Lamellum); LP (Lamellipodium); MR (Membrane ruffles). See also Appendix A.

**Figure 4 ijms-25-10705-f004:**
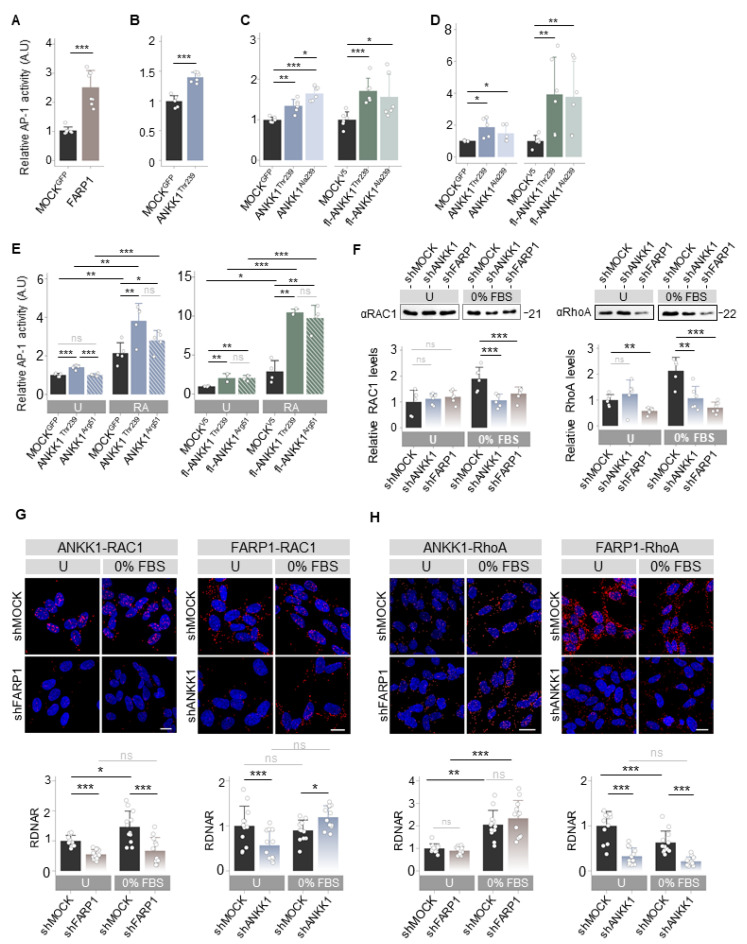
ANKK1 and FARP1 are components of the Wnt/PCP pathway activating RhoGTPases. (**A**–**E**). LRA of AP-1 dependent transcriptional activity for the Wnt/PCP pathway activation after transient overexpression of: (**A**–**C**) HEK293 cells transfected with (**A**) FARP1 (n = 8, 3 replicas each), (**B**) ANKK1 (n = 6, 3 replicas each), (**C**, blue) ANKK1^Thr239^ or ANKK1^Ala239^, and (**C**, green) fl-ANKK1^Thr239^ or fl-ANKK1^Ala239^ (n = 6, 3 replicas each), (**D**,**E**) proliferative SH-SY5Y cells transfected with (**D**, blue) ANKK1^Thr239^ or ANKK1^Ala239^, and (**D**, green) fl-ANKK1^Thr239^ or fl-ANKK1^Ala239^ (n = 6, 3 replicas each) and (**E**) undifferentiated (U) or RA treated (RA) SH-SY5Y cells transfected with (**E**, blue) ANKK1^Thr239^ or ANKK1^Arg51^, and (**E**, green) fl-ANKK1^Thr239^ or fl-ANKK1^Arg51^ (n = 5/4, 3 replicas each). (**F**). Western blot of RAC1 (left panel) and RhoA (right panel) in undifferentiated (U) or 48 h serum-starved (0% FBS) in shMOCK, shANKK1, and shFARP1 cell lines. For RAC1 and RhoA quantification total protein levels were assessed. Protein quantification is shown in the bar plots below (n = 6, 2 independent experiments). (**G**,**H**). PLA confocal images showing the interaction of endogenous: RAC1–ANKK1 (**G** left), RAC1–FARP1 (**G** right), RhoA–ANKK1 (**H** left) and RhoA–FARP1 (**H** right) in undifferentiated (U) or 48 h serum starved (0%FBS) shFARP1 and shANKK1 cell lines. Maximum intensity projections are shown. Scale bar: 10 µm. Analysis of RDNAR is shown in the bar plots below (n = 12 images, 3 independent experiments). Plots represent mean ± SD and individual values are displayed as dots. (**A**,**B**,**F**) One-sample *t-*test on the fold-change values was used for MOCK/U-group comparisons. (**C**,**D**,**G**,**H**) One-sample *t*-test on the fold-change values was used for MOCK-group comparisons. Student’s *t*-test on the fold-change values was used for inter-group comparisons. Bonferroni was used as a multiple-testing correction method. (**E**) One-sample *t*-test on the fold-change values was used for MOCK (U)-group comparisons. Student’s *t*-test (2 groups) or ANOVA (3 groups) on the fold-change values was used for inter-group comparisons. Bonferroni was used as a multiple-testing correction method. * *p* < 0.05; ** *p* < 0.01; *** *p* < 0.001; ns, not significant. Abbreviations: FBS: Fetal bovine serum; PLA: Proximity ligation assay; RA: Retinoic acid; RDNAR: Relative dots/nuclei area ratio; U: Undifferentiated. See also Appendix A.

**Figure 6 ijms-25-10705-f006:**
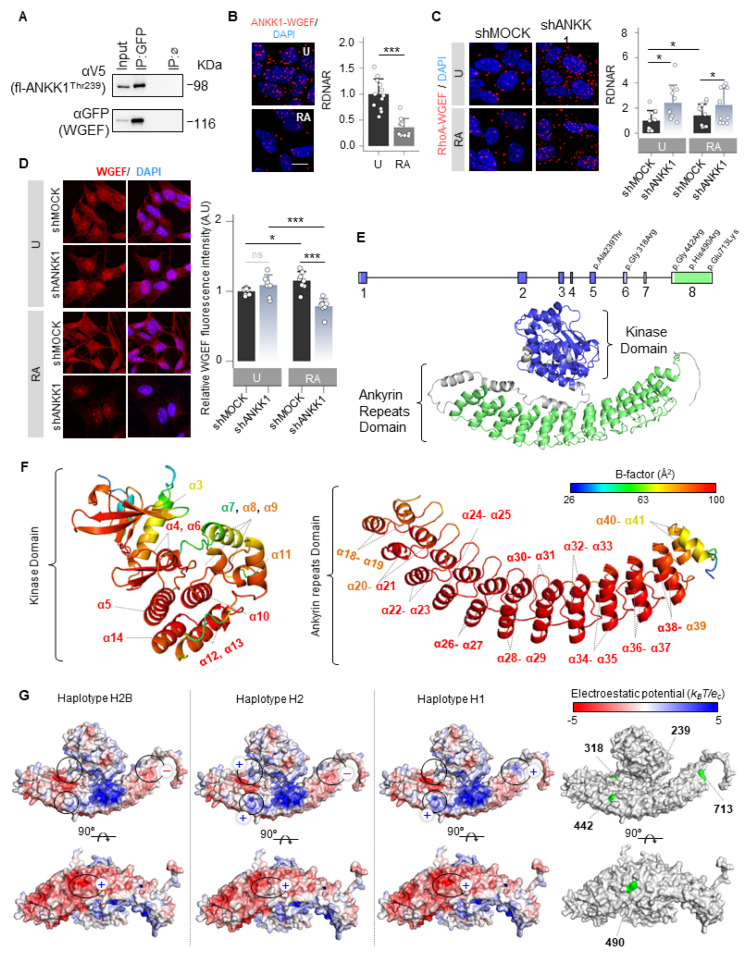
ANKK1 is a scaffold protein that interacts with Wnt/PCP components. (**A**). Co-IP of endogenous WGEF and ANKK1 in scMOCK and scfl-ANKK1^Thr239^ cell lines during proliferation. (**B**). ANKK1–WGEF interaction by PLA in undifferentiated (U) and RA-treated (RA) SH5YSY cells. Maximum intensity projections are shown. Scale bar: 10 µm. Analysis of RDNAR is shown in the bar plots below (n = 12 images, 2 independent experiments (**C**). PLA confocal images showing the endogenous RhoA–WGEF interaction in undifferentiated (U) and RA-treated (RA) shMOCK and shANKK1 knockdown cell lines. Maximum intensity projections are shown. Scale bar: 10 µm. Analysis of RDNAR is shown in the bar plots below (n = 11 images, 2 independent experiments). (**D**). Immunostaining of WGEF in undifferentiated (U) and RA-treated (RA) shMOCK and shANKK1 cell lines. Maximum intensity projections are shown. Scale bar: 15 µm. Analysis of WGEF intensity is shown in the adjacent bar plots (n = 8 images, 2 independent experiments). (**E**). ANKK1 gene and protein structures. Colors correspond to the kinase domain (blue; encoded by exons 1–6, comprising amino acid positions from 22 to 289) and to the ankyrin repeats domain (green; encoded by exon 8, comprising amino acid positions from 361 to 753, in green). (**F**). B-factor analysis for the ANKK1 protein. Dark blue represents the lowest B-factor values, while red represents the highest ones. (**G**). Electrostatic potential surfaces with solvent exclusion calculated by the APBS module implemented with PyMol for the different ANKK1 haplotypes: H2B, H2, and H1 (A red-white-blue color scheme with a range of −5/+5 is used, where red and blue colors correspond to negative and positive electrostatic potentials, respectively. To show different display options, each protein has been rotated 90°. Amino acid positions where the ANKK1 haplotypes/alleles differ (239, 318, 442, 490 and 713) are highlighted in green (right). Plots represent mean ± SD and individual values are displayed as dots. (**B**) One-sample *t*-test on the fold-change values was used for U-group comparison. (**C**) One-sample *t*-test on the fold-change values was used for shMOCK(U)-group comparison, and Bonferroni was used as a multiple-testing correction method. Student’s *t* test on the fold-change values was used for inter-group comparisons. * *p* < 0.05; *** *p* < 0.001; ns, not significant. Abbreviations: Co-IP: co-immunoprecipitation; IP:ø: non-specific IP; FBS: Fetal bovine serum; PLA: Proximity ligation assay; RA: Retinoic acid; RDNAR: Relative dots/nuclei area ratio; U: Undifferentiated; (+): Positively charged; (−): Negatively charged; (n): neutrally charged. See also Appendix A.

## Data Availability

Further information and requests for resources and reagents should be directed to and will be fulfilled by the Lead Contact, Janet Hoenicka (janet.hoenicka@sjd.es).

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
