# Peer review of "ANKK1 Is a Wnt/PCP Scaffold Protein for Neural F-ACTIN Assembly"

_ijms, 2024, doi:10.3390/ijms251910705_

Round 1

Reviewer 1 Report

Comments and Suggestions for Authors

Laura Domínguez-Berzosa et al. utilized cell lines such as SH-SY5Y and HEK293T and employed a range of experimental techniques, including co-immunoprecipitation (co-IP), proximity ligation assay (PLA), immunofluorescence staining, qRT-PCR, and a custom automatic image analysis method, to demonstrate the interaction between ANKK1 and FARP1 and its association with the Wnt/PCP pathway. These findings provide compelling evidence that ANKK1 serves as a scaffold protein for the Wnt/PCP pathway, contributing to the assembly of neuronal F-ACTIN, thereby elucidating its functional role in neuronal cells. While the study presents comprehensive data and offers valuable insights, several issues need to be addressed:

1: Improvement needed in the introduction. The introduction section requires further improvement. Particularly in the last paragraph, the rationale provided by the authors for hypothesizing the interaction between ANKK1 and FARP1 is not sufficiently substantiated.

2: Limitations of in vitro models. The manuscript predominantly relies on in vitro cell line models. It would be beneficial to include data from mouse models or other in vivo systems to validate the ANKK1-FARP1 interaction and support the involvement of ANKK1 in the Wnt/PCP signaling pathway.

3: Inconsistencies in citation numbering. There are inconsistencies in the citation numbering throughout the manuscript. For instance, the first sentence of the introduction cites reference number 32. The citations should be reorganized to follow a sequential order to enhance clarity and readability.

4: The English in the manuscript requires minor refinement.

Comments on the Quality of English Language

The English in the manuscript requires minor refinement.

Author Response

1: Improvement needed in the introduction. The introduction section requires further improvement. Particularly in the last paragraph, the rationale provided by the authors for hypothesizing the interaction between ANKK1 and FARP1 is not sufficiently substantiated.

We thank the reviewer for this comment. In this study, we found the ANKK1-FARP1 interaction using a hypothesis-free approach (based on an iTrack experiment). Following the reviewer's suggestion, we have modified the last paragraph and clarified this point. Furthermore, we added more information about FARP1 biology in neurons to support the hypothesis we propose in the last paragraph of the introduction.

We have modified the last paragraph accordingly (lines 70 – 75).

2: Limitations of in vitro models. The manuscript predominantly relies on in vitro cell line models. It would be beneficial to include data from mouse models or other in vivo systems to validate the ANKK1-FARP1 interaction and support the involvement of ANKK1 in the Wnt/PCP signaling pathway.

We thank the reviewer for this comment. In our research, we have thoroughly investigated the interaction between ANKK1 and FARP1 in cellular models. This novel finding reveals the biochemical function of ANKK1 and its association with known FARP1 functions in neurons. In the discussion section, we mentioned that our study has limitations, which we included in lines 475-478, where we highlight the need for further studies on the WNT/PCP pathway in ANKK1 knockout mouse models. These future studies are essential for gaining more insights into the role of ANKK1 proteins and their impact on behavior and susceptibility to addiction.  

3: Inconsistencies in citation numbering. There are inconsistencies in the citation numbering throughout the manuscript. For instance, the first sentence of the introduction cites reference number 32. The citations should be reorganized to follow a sequential order to enhance clarity and readability.

We appreciate this comment. We have corrected the citation numbering throughout the manuscript.

4: The English in the manuscript requires minor refinement.

We thank the reviewer for this comment, and we have read the article and made minor revisions (lines 179,191,242,243,287,305,342,365,448).

Reviewer 2 Report

Comments and Suggestions for Authors

In this manuscript, the role and the molecular mechanism of ANKK1 in neural F-ACTIN assembly were explored. The authors found that ANKK1 interacts with the synapse protein FARP1, regulating its binding to the RhoGTPases RAC1 and RhoA. ANKK1FARP1 colocalized in F-ACTIN-rich structures for neuronal maturation and migration, and both proteins activate the Wnt/PCP pathway. Both ANKK1 and FARP1 involve in neuritic spine outgrowth, and ANKK1 promotes neuritogenesis, Knockdown of ANKK1 or FARP1 affects RhoGTPases expression and neural differentiation. During neuronal differentiation, ANKK1-WGEF interaction is downregulated, while ANKK1FARP1 interaction is increased, suggesting that ANKK1 recruits Wnt/PCP components for bidirectional control of F-ACTIN assembly. A lot of work were done in this project.

However, there are minor problems remained in this study.

1.     The Co-IP assay was not shown probably, negative control was missing in Figures 1 and 6, although the original-images looks complete.

2.     Line 115, “D” is wrong.

Author Response

In this manuscript, the role and the molecular mechanism of ANKK1 in neural F-ACTIN assembly were explored. The authors found that ANKK1 interacts with the synapse protein FARP1, regulating its binding to the RhoGTPases RAC1 and RhoA. ANKK1–FARP1 colocalized in F-ACTIN-rich structures for neuronal maturation and migration, and both proteins activate the Wnt/PCP pathway. Both ANKK1 and FARP1 involve in neuritic spine outgrowth, and ANKK1 promotes neuritogenesis, Knockdown of ANKK1 or FARP1 affects RhoGTPases expression and neural differentiation. During neuronal differentiation, ANKK1-WGEF interaction is downregulated, while ANKK1–FARP1 interaction is increased, suggesting that ANKK1 recruits Wnt/PCP components for bidirectional control of F-ACTIN assembly. A lot of work were done in this project.

However, there are minor problems remained in this study.

  1. The Co-IP assay was not shown probably, negative control was missing in Figures 1 and 6, although the original-images looks complete. Response: 

    We thank the reviewer for this comment. We have enlarged the font so that it is not confusing. In the coIP presented in Figures 1 and 6, we show both the total lysate of each protein (input) and the coIP of the protein of interest (FARP1 or WGEF). The lane labeled as 'non-specific IP; IP:Æ' represents the negative control for the coIP. In this control, an antibody that does not target any protein in the cell lysate (e.g., anti-V5 or anti-HA) was used, ensuring that no specific interaction occurs. This confirms that the observed interactions in the experimental samples are specific to the antibody used for the protein of interest.

  2. Line 115, “D” is wrong. Response: 

    We thank the reviewer for this annotation. We have corrected all the figure legend.  We highlighted all changes (lines, 118, 120-122).